# Asynchronous Coordinate Descent under More Realistic Assumption*

**Tao Sun**
National University of Defense Technology
Changsha, Hunan 410073, China
nudtsuntao@163.com

**Robert Hannah**
University of California, Los Angeles
Los Angeles, CA 90095, USA
RobertHannah89@math.ucla.edu

**Wotao Yin**
University of California, Los Angeles
Los Angeles, CA 90095, USA
wotaoyin@math.ucla.edu

## Abstract

Asynchronous-parallel algorithms have the potential to vastly speed up algorithms by eliminating costly synchronization. However, our understanding of these algorithms is limited because the current convergence theory of asynchronous block coordinate descent algorithms is based on somewhat unrealistic assumptions. In particular, the age of the shared optimization variables being used to update blocks is assumed to be independent of the block being updated. Additionally, it is assumed that the updates are applied to randomly chosen blocks.

In this paper, we argue that these assumptions either fail to hold or will imply less efficient implementations. We then prove the convergence of asynchronous-parallel block coordinate descent under more realistic assumptions, in particular, always without the independence assumption. The analysis permits both the deterministic (essentially) cyclic and random rules for block choices. Because a bound on the asynchronous delays may or may not be available, we establish convergence for both bounded delays and unbounded delays. The analysis also covers nonconvex, weakly convex, and strongly convex functions. The convergence theory involves a Lyapunov function that directly incorporates both objective progress and delays. A continuous-time ODE is provided to motivate the construction at a high level.

## 1   Introduction

In this paper, we consider the asynchronous-parallel block coordinate descent (async-BCD) algorithm for solving the unconstrained minimization problem

$$\min_{x \in \mathbb{R}^N} f(x) = f(x_1, \ldots, x_N), \tag{1}$$

where $f$ is a differentiable function and $\nabla f$ is $L$-Lipschitz continuous. Async-BCD [14, 13, 16] has virtually the same implementation as regular BCD. The difference is that the threads doing the parallel computation do not wait for all others to finish and share their updates before starting the next iteration, but merely continue to update with the most recent solution-vector information available[2].

In traditional algorithms, latency, bandwidth limits, and unexpected drains on resources, that delay the update of even a single thread will cause the entire system to wait. By eliminating this costly idle time, asynchronous algorithms can be much faster than traditional ones.

In async-BCD, each agent continually updates the solution vector, one block at a time, leaving all other blocks unchanged. Each block update is a read-compute-update cycle. It begins with an agent reading $x$ from shared memory or a parameter server, and saving it in a local cache as $\hat{x}$. The agent then computes $-\frac{1}{L}\nabla_i f(\hat{x})$, a block partial gradient[3]. The final step of the cycle depends on the parallel system setup. In a shared memory setup, the agent reads block $x_i$ again and writes $x_i - \frac{\gamma_k}{L}\nabla_i f(\hat{x})$ to $x_i$ (where $\gamma^k$ is the step size). In parameter-server setup, the agent can send $-\frac{1}{L}\nabla_i f(\hat{x})$ and let the server update $x_i$. Other setups are possible, too. The iteration counter $k$ increments upon the completion of any block update, and the updating block is denoted as $i_k$.

Many iterations may occur between the time a computing node reads the solution vector $\hat{x}$ into memory, and the time that the node's corresponding update is applied to the shared solution vector. Because of this, the iteration of asyn-BCD is, therefore, modeled [14] as

$$x_{i_k}^{k+1} = x_{i_k}^k - \frac{\gamma_k}{L}\nabla_{i_k} f(\hat{x}^k), \tag{2}$$

where $\hat{x}^k$ is a potentially outdated version of $x^k$, and $x_j^{k+1} = x_j^k$ for all non-updating blocks $j \neq i_k$. The convergence behavior of this algorithm depends on the sequence of updated blocks $i_k$, the step size sequence $\gamma_k$, as well as the ages of the blocks of $\hat{x}^k$ relative to $x^k$. We define the *delay vector* $\vec{j}(k) = (j(k,1), j(k,2), \ldots, j(k,N)) \in \mathbb{Z}^N$, which represents the how outdated each of the blocks are. Specifically, we have define:

$$\hat{x}^k = (x_1^{k-j(k,1)}, x_2^{k-j(k,2)}, \ldots, x_N^{k-j(k,N)}). \tag{3}$$

The *k'th delay* (or *current delay*) is $j(k) = \max_{1 \leq i \leq N}\{j(k,i)\}$.

## 1.1 Dependence between delays and blocks

In previous analyses [13, 14, 16, 9], it is assumed that the block index $i_k$ and the delay $\vec{j}(k)$ were independent sequences. This simplifies proofs, for example, giving $\mathbb{E}_{i_k}(P_{i_k}\nabla f(\hat{x}^k)) = \frac{1}{N}\nabla f(\hat{x}^k)$ when $i_k$ is chosen at random, where $P_i$ denotes the projection to the $i$th block. Without independence, $\vec{j}(k)$ will depend on $i_k$, causing the distribution of $\hat{x}^k$ to be different for each possible $i_k$, thus breaking the previous equality. However, the independence assumption is unrealistic in practice.

Consider a problem where some blocks are more expensive to update than others[4]. Blocks that take longer to update should have greater delays when they are updated because more other updates will have occurred between the time that $\hat{x}$ is read and when the update is applied. For the same reason, updates on blocks assigned to slower or busier agents will generally have greater delays. Indeed this turns out to be the case in practice. Experiments were performed on a cluster with 2 nodes, each with 16 threads running on an Intel Xeon CPU E5-2690 v2. The algorithm was applied to the logistic regression problem on the "news20" data set from LIBSVM, with 64 contiguous coordinate blocks of equal size. Over 2000 epochs, blocks 0, 1, and 15 had average delays of 351, 115, and 28, respectively. ASync-BCD completed this over 7x faster than the corresponding synchronous algorithm using the same computing resources, with a nearly equal decrease in objective function. Even when blocks have balanced difficulty, and the computing nodes have equal computing power, this dependence persists. We assigned 20 threads to each core, with each thread assigned to a block of 40 coordinates with an equal numbers of nonzeros. The mean delay varied from 29 to 50 over the threads. This may be due to the cluster scheduler or issues of data locality, which were hard to examine. Clearly, there is strong dependence of the delays $\vec{j}(k)$ on the updated block $i_k$.

## 1.2 Stochastic and deterministic block rules

This paper considers two different *block rules*: *deterministic* and *stochastic*. For the stochastic block rule, at each update, a block is chosen from $\{1, 2, \ldots, N\}$ uniformly at random[5], for instance in

[14, 13, 16]. For the deterministic rule, $i_k$ is an arbitrary sequence that is assumed to be *essentially cyclic*. That is, there is an $N' \in \mathbb{N}$, $N' \geq N$, such that each block $i \in \{1, 2, \ldots, N\}$ is updated at least once in a window of $N'$, that is,

For each $t \in \mathbb{Z}^+$, $\exists$ integer $K(i,t) \in \{tN', tN'+1, \ldots, (1+t)N'-1\}$ such that $i_{K(i,t)} = i$.

This encompasses different kinds of *cyclic* rules such as fixed ordering, random permutation, and greedy selection. The stochastic block rule is easier to analyze because taking expectation will yield a good approximation to the full gradient. It ensures the every block is updated at the specified frequency. However, it can be expensive or even infeasible to implement for the following reasons. In the shared memory setup, stochastic block rules require random data access, which is not only significantly slower than sequential data access but also cause frequent *cache misses* (waiting for data being fetched from slower cache or the main memory). The cyclic rules clearly avoid these issues since data requirements are predictable. In the parameter-server setup where workers update randomly assigned blocks at each step, each worker must either store all the problem data necessary to update any block (which may mean massive storage requirements) or read the required data from the server at every step (which may mean massive bandwidth requirements). Clearly this permanently assigning blocks to agents avoids these issues. On the other hand, the analysis of cyclic rules generally has to consider the worst-case ordering and necessarily gives worse performance in the worst case[19]. In practice, worst-case behavior is rare, and cyclic rules often lead to good performance [7, 8, 3].

## 1.3 Bounded and unbounded delays

We consider different delay assumptions as well. *Bounded delay* is when $j(k) \leq \tau$ for some fixed $\tau \in \mathbb{Z}^+$ and all iterations $k$; while the *unbounded delay* allows $\sup_k \{j(k)\} = +\infty$. Bounded and unbounded delays can be further divided into deterministic and stochastic. Deterministic delays refer to a sequence of delay vectors $\vec{j}(0), \vec{j}(1), \vec{j}(2), \ldots$ that is arbitrary or follows an unknown distribution so is treated as arbitrary. Our stochastic delay results apply to distributions that decay faster than $O(k^{-3})$. Deterministic unbounded delays apply to the case when async-BCD runs on unfamiliar hardware platforms. For convergence, we require a finite $\liminf_k \{j(k)\}$ and the current step size $\eta^k$ to be adaptively chosen to the current delay $j(k)$, which must be measured or overestimated. Bounded delays and stochastic unbounded delays apply when the user can provide a bound or delay distribution, respectively. The user can obtain these from previous experience or by running a pilot test. In return, a fixed step size allows convergence, and measuring the current delay is not needed.

## 1.4 Contributions

Our contributions are mainly convergence results for three kinds of delays: bounded, stochastic unbounded, deterministic unbounded, that are obtained without the artificial independence between the block index and the delay. The results are provided for nonconvex, convex, and strongly convex functions with Lipschitz gradients. Sublinear rates and linear rates are provided, which match the rates for the corresponding synchronous algorithms in terms of order of magnitude. Due to space limitation, we restrict ourselves to Lipschitz differentiable functions and leave out nonsmooth proximable functions. Like many analyses of asynchronous algorithms, our proofs are built on the construction of Lyapunov functions. We provide a simple ODE-based (i.e., continuous time) construction for bounded delays to motivate the construction of the Lyapunov function in the standard discrete setting. Our analysis brings great news to the practitioner. Roughly speaking, in a variety of setting, even when there is no load balancing (thus the delays may depend on the block index) or bound on the delays, convergence of async-BCD can be assured by using our provided step sizes.

Our proofs do not treat asynchronicity as noise, as many papers do[6], because modelling delays in this way appears to destroy valuable information, and leads to inequalities that are too blunt to obtain stronger results. This is why sublinear and linear rates can be established for weak and strong convex problems respectively, even when delays depend on the blocks and are potentially unbounded. Our main focus was to prove new convergence results in a new setting, not to obtain the best possible rates. Space limitations make this difficult, and we leave it for future work. The main message is that even without the independence assumption, convergence of the same order as for the corresponding synchronous algorithm occurs. The step sizes and rates obtained may be overly pessimistic for the

practitioner to use. In practice, we find that using the standard synchronous step size results in convergence, and the observed rate of convergence is extremely close to that of the synchronous counterpart. With the independence assumption, convergence rates for asynchronous algorithms have recently been proven to be asymptotically the same as their synchronous counterparts[10].

## 1.5 Related work

Our work extends the theory on asynchronous BCD algorithms such as [18, 14, 13]. However, their analyses rely on the independence assumption and assume bounded delays. The bounded delay assumption was weakened by recent papers [9, 17], but independence and random blocks were still needed. Recently [12] proposes (in the SGD setting) an innovative "read after" sequence relabeling technique to create the independence. However, enforcing independence in this way creates other artificial implementation requirements that may be problematic: For instances, agents must read "all shared data parameters and historical gradients before starting iterations", even if not all of this is required to compute updates. Our analysis does not require these kinds of implementation fixes. Also, our analysis also works for unbounded delays and deterministic block choices.

Related recent works also include [1, 2], which solve our problem with additional convex block-separable terms in the objective. In the first paper [1], independence between blocks and delays is avoided. However, they require a step size that diminishes at $1/k$ and that the sequence of iterate is bounded (which in general may not be true). The second paper [2] relaxes independence by using a different set of assumptions. In particular, their assumption D3 assumes that, regardless of the previous updates, there is a universally positive chance for every block to be updated in the next step. This Markov-type assumption relaxes the independence assumption but does not avoid it. Paper [15] addressed this issue by decoupling the parameters read by each core from the virtual parameters on which progress is actually defined. Based on the idea of [16], [12] addressed the dependence problem in related work. In the convex case with a bounded delay $\tau$, the step size in paper [14] is $O(\frac{1}{\tau^2/N})$. In their proofs, the Lyapunov function is based on $\|x^k - x^*\|_2^2$. Our analysis uses a Lyapunov function consisting of both the function value and the sequence history, where the latter vanishes when delays vanish. If the $\tau$ is much larger than the blocks of the problem, our step size $O(\frac{1}{\tau})$ is better even under our much weaker conditions. The step size bound in [16, 9, 4] is $O(\frac{1}{1+2\tau/\sqrt{N}})$, which is better than ours, but they need the independence assumption and the stochastic block rule. Recently, [20] introduces an asynchronous primal-dual method for a problem similar to ours but having additional affine linear constraints. The analysis assumes bounded delays, random blocks, and independence.

## 1.6 Notation

We let $x^*$ denote any minimizer of $f$. For the update in (2), we use the following notation:

$$\Delta^k := x^{k+1} - x^k \overset{(2)}{=} -\frac{\gamma_k}{L}\nabla_{i_k}f(\hat{x}^k), \qquad d^k := x^k - \hat{x}^k. \tag{4}$$

We also use the convention $\Delta^k := 0$ if $k < 0$. Let $\chi^k$ be the sigma algebra generated by $\{x^0, x^1, \ldots, x^k\}$. Let $\mathbb{E}_{\vec{j}(k)}$ denote the expectation over the value of $\vec{j}(k)$ (when it is a random variable). $\mathbb{E}$ denotes the expectation over all random variables.

## 2 Bounded delays

In this part, we present convergence results for the bounded delays. If the gradient of the function is $L$-Lipschitz (even if the function is nonconvex), we present the convergence for both the deterministic and stochastic block rule. If the function is convex, we can obtain a sublinear convergence rate. Further, if the function is restricted strongly convex, a linear convergence rate is obtained.

### 2.1 Continuous-time analysis

Let $t$ be time in this subsection. Consider the ODE

$$\dot{x}(t) = -\eta\nabla f(\hat{x}(t)), \tag{5}$$

where $\eta > 0$. If we set $\hat{x}(t) \equiv x(t)$, this system describes a gradient flow, which monotonically decreases $f(x(t))$, and its discretization is the gradient descent iteration. Indeed, we have

$\frac{d}{dt}f(x(t)) = \langle\nabla f(x(t)), \dot{x}(t)\rangle \overset{(5)}{=} -\frac{1}{\eta}\|\dot{x}(t)\|_2^2$. Instead, we allow delays (i.e., $\hat{x}(t) \neq x(t)$) and impose the bound $c > 0$ on the delays:

$$\|\hat{x}(t) - x(t)\|_2 \leq \int_{t-c}^t \|\dot{x}(s)\|_2 ds. \tag{6}$$

The delays introduce inexactness to the gradient flow $f(x(t))$. We lose monotonicity. Indeed,

$$\frac{d}{dt}f(x(t)) = \langle\nabla f(x(t)), \dot{x}(t)\rangle = \langle\nabla f(\hat{x}(t)), \dot{x}(t)\rangle + \langle\nabla f(x(t)) - \nabla f(\hat{x}(t)), \dot{x}(t)\rangle \tag{7}$$

$$\overset{a)}{\leq} -\frac{1}{\eta}\|\dot{x}(t)\|_2^2 + L\|x(t) - \hat{x}(t)\|_2 \cdot \|\dot{x}(t)\|_2 \overset{b)}{\leq} -\frac{1}{2\eta}\|\dot{x}(t)\|_2^2 + \frac{\eta c L^2}{2}\int_{t-c}^t \|\dot{x}(s)\|_2^2 ds,$$

Here a) is from (5) and Lipschitzness of $\nabla f$ and b) is from the Cauchy-Schwarz inequality $L\|x(t) - \hat{x}(t)\|_2 \cdot \|\dot{x}(t)\|_2 \leq \frac{\|\dot{x}(t)\|_2^2}{2\eta} + \frac{\eta L^2\|x(t)-\hat{x}(t)\|_2^2}{2}$ and $\|x(t)-\hat{x}(t)\|_2^2 \overset{(6)}{\leq} c\int_{t-c}^t\|\dot{x}(s)\|_2^2 ds$. The inequalities are generally unavoidable. Therefore, we design an energy function with both $f$ and a weighted total kinetic term, where $\gamma > 0$ will be decided below:

$$\xi(t) = f(x(t)) + \gamma\int_{t-c}^t (s - (t-c))\|\dot{x}(s)\|_2^2 ds. \tag{8}$$

By substituting the bound on $\frac{d}{dt}f(x(t))$ in (7), we get the time derivative:

$$\dot{\xi}(t) = \frac{d}{dt}f(x(t)) + \gamma c\|\dot{x}(t)\|_2^2 - \gamma\int_{t-c}^t \|\dot{x}(s)\|_2^2 ds$$

$$\leq -(\frac{1}{2\eta} - \gamma c)\|\dot{x}(t)\|_2^2 - (\gamma - \frac{\eta c L^2}{2})\int_{t-c}^t \|\dot{x}(s)\|_2^2 ds \tag{9}$$

As long as $\eta < \frac{1}{Lc}$, there exists $\gamma > 0$ such that $(\frac{1}{2\eta} - \gamma c) > 0$ and $(\gamma - \frac{\eta c L^2}{2}) > 0$, so $\xi(t)$ is monotonically nonincreasing. Assume $\min f$ is finite. Since $\xi(t)$ is lower bounded by $\min f$, $\xi(t)$ must converge, hence $\dot{\xi} \to 0$, $\dot{x}(t) \to 0$ by (8). $\nabla f(\hat{x}(t)) \to 0$ by (5), and $\hat{x}(t) - x(t) \to 0$ by (6). The last two results further yield $\nabla f(x(t)) \to 0$.

## 2.2 Discrete analysis

The analysis for our discrete iteration (2) is based on the following Lyapunov function:

$$\xi_k := f(x^k) + \frac{L}{2\varepsilon}\sum_{i=k-\tau}^{k-1}(i - (k-\tau) + 1)\|\Delta^i\|_2^2. \tag{10}$$

for some $\varepsilon > 0$ determined later based on the step size and $\tau$, the bound on the delays. The constant $\varepsilon$ is not an algorithm parameter. In the lemma below, we present a fundamental inequality, which states, regardless of which block $i_k$ is updated and which $\hat{x}^k$ is used to compute the update in (2), there is a sufficient descent in our Lyapunov function.

**Lemma 1 (sufficient descent for bounded delays)** *Conditions: Let $f$ be a function (possibly nonconvex) with $L$-Lipschitz gradient and finite $\min f$. Let $(x^k)_{k\geq 0}$ be generated by the async-BCD algorithm (2), and the delays be bounded by $\tau$. Choose the step size $\gamma_k \equiv \gamma = \frac{2c}{2\tau+1}$ for arbitrary fixed $0 < c < 1$. Result: we can choose $\varepsilon > 0$ to obtain*

$$\xi_k - \xi_{k+1} \geq \frac{1}{2}(\frac{1}{\gamma} - \frac{1}{2} - \tau)L \cdot \|\Delta^k\|_2^2, \tag{11}$$

*Consequently,*

$$\lim_k \|\Delta^k\|_2 = 0 \qquad (12) \qquad and \qquad \min_{1\leq i\leq k}\|\Delta^i\|_2 = o(1/\sqrt{k}). \tag{13}$$

So we have that the smallest gradient obtained by step $k$ decays faster than $O(1/\sqrt{k})$. Based on the lemma, we obtain a very general result for nonconvex problems.

**Theorem 1** *Assume the conditions of Lemma 1, for $f$ that may be nonconvex. Under the deterministic block rule, we have*

$$\lim_k \|\nabla f(x^k)\|_2 = 0, \quad \min_{1 \le i \le k} \|\nabla f(x^k)\|_2 = o(1/\sqrt{k}). \tag{14}$$

This rate has the same order of magnitude as standard gradient descent for a nonconvex function.

## 2.3 Stochastic block rule

Under the stochastic block rule, an agent picks a block from $\{1, 2, ..., N\}$ uniformly randomly at the beginning of each update. For the $k$th completed update, the index of the chosen block is $i_k$. Our result in this subsection relies on the following assumption on the random variable $i_k$:

$$\mathbb{E}_{i_k}(\|\nabla_{i_k} f(x^{k-\tau})\|_2 \mid \chi^{k-\tau}) = \frac{1}{N} \sum_{i=1}^N \|\nabla_i f(x^{k-\tau})\|_2, \tag{15}$$

where $\chi^k = \sigma(x^0, x^1, \ldots, x^k, \vec{j}(0), \vec{j}(1), \ldots, \vec{j}(k))$, $k = 0, 1, \ldots$, is the filtration that represents the information that is accumulated as our algorithm runs. It is important to note that (15) uses $x^{k-\tau}$ instead of $\hat{x}^k$ because $\hat{x}^k$ may depend on $i_k$. This condition essentially states that, given the information at iteration $k - \tau$ and earlier, $i_k$ is uniform at step $k$. *We can relax* (15) *to nearly-uniform distributions*. Indeed, Theorem 2 below only needs that every block has a nonzero probability of being updated given $\chi^{k-\tau}$, that is,

$$\mathbb{E}(\|\nabla_{i_k} f(x^{k-\tau})\|_2 \mid \chi^{k-\tau}) \ge \frac{\bar{\varepsilon}}{N} \sum_{i=1}^N \|\nabla_i f(x^{k-\tau})\|_2, \tag{16}$$

for some universal $\bar{\varepsilon} > 0$. The interpretation is that though $i_k$ and $\nabla f(x^{k-\tau})$ are dependent, since $\tau$ iterations have passed, $\nabla f(x^{k-\tau})$ has a limited influence on the distribution $i_k$: There is a minimum probability that each index is chosen given sufficient time. For convenience and simplicity, we assume (15) instead of (16) .

Next, we present a general result for a possibly nonconvex objective $f$.

**Theorem 2** *Assume the conditions of Lemma 1. Under the stochastic block rule and assumption* (15)*, we have:*

$$\lim_k \mathbb{E}\|\nabla f(x^k)\|_2 = 0, \quad \min_{1 \le i \le k} \mathbb{E}\|\nabla f(x^k)\|_2^2 = o(1/k). \tag{17}$$

### 2.3.1 Sublinear rate under convexity

When the function $f$ is convex, we can obtain convergence rates, for which we need a slightly modified Lyapunov function

$$F_k := f(x^k) + \delta \cdot \sum_{i=k-\tau}^{k-1} (i - (k - \tau) + 1) \|\Delta^i\|_2^2, \tag{18}$$

where $\delta := [1 + \frac{\varepsilon}{2\tau}(\frac{1}{\gamma} - \frac{1}{2} - \tau)] \frac{L}{2\varepsilon}$. Here, we assume $\tau \ge 1$. Since $\tau$ is just an upper bound of the delays, the delays can be 0. We also define $\pi_k := \mathbb{E}(F_k - \min f), \quad S(k, \tau) := \sum_{i=k-\tau}^{k-1} \|\Delta^i\|_2^2$.

**Lemma 2** *Assume the conditions of Lemma 1. Furthermore, let $f$ be convex and use the stochastic block rule. Let $\overline{x^k}$ denote the projection of $x^k$ to $\arg\min f$, assumed to exist, and let $\beta := \max\{\frac{8NL^2}{\gamma^2}, (12N + 2)L^2\tau + \delta\tau\}, \quad \alpha := \beta/[\frac{L}{4\tau}(\frac{1}{\gamma} - \frac{1}{2} - \tau)]$. Then we have:*

$$(\pi_k)^2 \le \alpha(\pi_k - \pi_{k+1}) \cdot (\delta\tau \mathbb{E}S(k, \tau) + \mathbb{E}\|x^k - \overline{x^k}\|_2^2). \tag{19}$$

When $\tau = 1$ (nearly no delay), we can obtain $\beta = O(NL^2/\gamma^2)$ and $\alpha = O(\beta\gamma/L) = O(NL/\gamma)$, which matches the result of standard BCD. This is used to prove sublinear convergence.

**Theorem 3** *Assume the conditions of Lemma 1. Furthermore, let $f$ be convex and coercive[7], and use the stochastic block rule. Then we have:*

$$\mathbb{E}(f(x^k) - \min f) = O(1/k). \tag{20}$$

### 2.3.2 Linear rate under convexity

We next consider when $f$ is $\nu$-restricted strongly convex[8] in addition to having $L$-Lipschitz gradient. That is, for $x \in \mathrm{dom}(f)$, $\langle \nabla f(x), x - \mathrm{Proj}_{\mathrm{argmin}\, f}(x) \rangle \geq \nu \cdot \mathrm{dist}^2(x, \mathrm{argmin}\, f)$.

**Theorem 4** *Assume the conditions of Lemma 1. Furthermore, let $f$ be $\nu$-strongly convex, and use the stochastic block rule. Then we have:*

$$\mathbb{E}(f(x^k) - \min f) = O(c^k), \tag{21}$$

*where $c := \frac{\alpha}{\min\{\nu,1\}} / (1 + \frac{\alpha}{\min\{\nu,1\}}) < 1$ for $\alpha$ given in Lemma 2.*

## 3 Stochastic unbounded delay

In this part, the delay vector $\vec{j}(k)$ is allowed to be an unbounded random variable. Under some mild restrictions on the distribution of $\vec{j}(k)$, we can still establish convergence. In light of our continuous-time analysis, we must develop a new bound for the last inner product in (7), which requires the tail distribution of $j(k)$ to decay sufficiently fast. Specifically, we define a sequence of fixed parameters $p_j$ such that $p_j \geq \mathbb{P}(j(k) = j), \forall k$, $s_l = \sum_{j=l}^{+\infty} j p_j$, and $c_i := \sum_{l=i}^{+\infty} s_l$. Clearly, $c_0$ is larger than $c_1, c_2, \ldots$, and we need $c_0$ to be finite. Distributions with $p_j = \mathcal{O}(j^{-t})$, for $t > 3$, and exponential-decay distributions satisfy this requirement. Define the Lyapunov function $G_k$ as $G_k := f(x^k) + \bar{\delta} \cdot \sum_{i=0}^{k-1} c_{k-1-i} \|\Delta^i\|_2^2$, where $\bar{\delta} := \frac{L}{2\varepsilon} + (\frac{1}{\gamma} - \frac{1}{2})\frac{L}{c_0} - \frac{L}{\sqrt{c_0}}$. To simplify the presentation, we define $R(k) := \sum_{i=0}^{k} c_{k-i} \mathbb{E}\|\Delta^i\|_2^2$.

**Lemma 3 (Sufficient descent for stochastic unbounded delays)** *Conditions: Let $f$ be a function (which may be nonconvex) with $L$-Lipschitz gradient and finite $\min f$. Let delays be stochastic unbounded. Use step size $\gamma_k \equiv \gamma = \frac{2c}{2\sqrt{c_0}+1}$ for arbitrary fixed $0 < c < 1$. Results: we can set $\varepsilon > 0$ to ensures sufficient descent:*

$$\mathbb{E}[G_k - G_{k+1}] \geq \frac{L}{c_0}(\frac{1}{\gamma} - \frac{1}{2} - \sqrt{c_0})R(k). \tag{22}$$

*And we have*

$$\lim_k \mathbb{E}\|\Delta^k\|_2 = 0 \quad and \quad \lim_k \mathbb{E}\|d^k\|_2 = 0. \tag{23}$$

### 3.1 Deterministic block rule

**Theorem 5** *Let the conditions of Lemma 3 hold for $f$. Under the deterministic block rule (§1.2), we have:*

$$\lim_k \mathbb{E}\|\nabla f(x^k)\|_2 = 0. \tag{24}$$

### 3.2 Stochastic block rule

Recall that under the stochastic block rule, the block to update is selected uniformly at random from $\{1, 2, \ldots, N\}$. The previous assumption (15), which is made for bounded delays, need to be updated into the following assumption for unbounded delays:

$$\mathbb{E}_{i_k}(\|\nabla_{i_k} f(x^{k-j(k)})\|_2^2) = \frac{1}{N}\sum_{i=1}^{N} \|\nabla_i f(x^{k-j(k)})\|_2^2, \tag{25}$$

where $j(k)$ is still a variable on both sides. As argued below (15), the uniform distribution can easily be relaxed to a nearly-uniform distribution, but we use the former for simplicity.

**Theorem 6** *Let the conditions of Lemma 3 hold. Under the stochastic block rule and assumption (25), we have*

$$\lim_k \mathbb{E}\|\nabla f(x^k)\|_2 = 0. \tag{26}$$

### 3.2.1 Convergence rate

When $f$ is convex, we can derive convergence rates for $\phi_k := \mathbb{E}(G_k - \min f)$.

**Lemma 4** *Let the conditions of Lemma 3 hold, and let $f$ be convex. Let $\overline{x^k}$ denote the projection of $x^k$ to $\operatorname{argmin} f$. Let $\overline{\beta} = \max\{\frac{8NL^2}{\gamma^2 c_0}, (12N+2)L^2 + \overline{\delta}\}$ and $\overline{\alpha} = \overline{\beta}/[\frac{L}{2}(\frac{1}{\gamma} - \frac{1}{2} - \sqrt{c_0})]$. Then we have*

$$(\phi_k)^2 \leq \overline{\alpha}(\phi_k - \phi_{k+1}) \cdot (\overline{\delta}R(k) + \mathbb{E}\|x^k - \overline{x^k}\|_2^2), \tag{27}$$

A sublinear convergence rate can be obtained if $\sup_k\{\mathbb{E}\|x^k - \overline{x^k}\|_2^2\} < +\infty$, which can be ensured by adding a projection to a large artificial box set that surely contains the solution. Here we only present a linear convergence result.

**Theorem 7** *Let the conditions of Lemma 3 hold. In addition, let $f$ be $\nu$-restricted strongly convex and set step size $\gamma_k \equiv \gamma < \frac{2}{2\sqrt{c_0}+1}$, with $c = \frac{\overline{\alpha}\max\{1,\frac{1}{\nu}\}}{1+\overline{\alpha}\max\{1,\frac{1}{\nu}\}} < 1$. Then,*

$$\mathbb{E}(f(x^k) - \min f) = O(c^k). \tag{28}$$

## 4   Deterministic unbounded delays

In this part, we consider deterministic unbounded delays, which require delay-adaptive step sizes. Set positive sequence $(\epsilon_i)_{i \geq 0}$ (which can be optimized later given actual delays) such that $\kappa_i := \sum_{j=i}^{+\infty}\epsilon_j$ obeys $\kappa_1 < +\infty$. Set $D_j := \frac{1}{2} + \frac{\kappa_1}{2} + \sum_{i=1}^{j}\frac{1}{2\epsilon_i}$. We use a new Lyapunov function $H_k := f(x^k) + \frac{L}{2}\sum_{i=1}^{+\infty}\kappa_i\|\Delta^{k-i}\|_2^2$. Let $T \geq \liminf j(k)$, and let $Q_T$ be the subsequence of $\mathbb{N}$ where the current delay is less than $T$. We prove convergence on the family of subsequences $x^k$, $k \in Q_T$. The algorithm is independent of the choice of $T$. The algorithm is run as before, and after completion, an arbitrarily large $T \geq \liminf j(k)$ can be chosen. Extending the result to standard sequence convergence has proven intractable.

**Lemma 5 (sufficient descent for unbounded deterministic delays)** *Conditions: Let $f$ be a function (which may be nonconvex) with $L$-Lipschitz gradient and finite $\min f$. The delays $j(k)$ are deterministic and obey $\liminf j(k) < \infty$. Use step size $\gamma_k = c/D_{j(k)}$ for arbitrary fixed $0 < c < 1$. Results: We have*

$$H_k - H_{k+1} \geq L(\frac{1}{\gamma_k} - D_{j(k)})\|\Delta^k\|_2^2, \qquad \lim_k \|\Delta^k\|_2 = 0. \tag{29}$$

*On any subsequence $Q_T$ (for arbitrarily large $T \geq \liminf j(k)$), we have:*

$$\lim_{(k \in Q_T) \to \infty}\|d^k\|_2 = 0, \qquad \lim_{(k \in Q_T) \to \infty}\|\nabla_{i_k}f(\hat{x}^k)\|_2 = 0,$$

To prove our next result, we need a new assumption: essentially cyclically semi-unbounded delay (ECSD), which is slightly stronger than the essentially cyclic assumption. In every window of $N'$ steps, every index $i$ is updated at least once with a delay less than $B$ (at iteration $K(i,t)$). The number $B$ just needs to exist and can be arbitrarily large. It does not affect the step size.

**Theorem 8** *Let the conditions of Lemma 5 hold. For the deterministic index rule under the ECSD assumption, for $T \geq B$, we have:*

$$\lim_{(k \in Q_T) \to \infty}\|\nabla f(x^k)\|_2 = 0. \tag{30}$$

## 5   Conclusion

In summary, we have proven a selection of convergence results for async-BCD under bounded and unbounded delays, and stochastic and deterministic block choices. These results do not require the independence assumption that occurs in the vast majority of other work so far. Therefore they better model the behavior of real asynchronous solvers. These results were obtained with the use of Lyapunov function techniques, and treating delays directly, rather than modelling them as noise. Future work may involve obtaining a more exhaustive list of convergence results, sharper convergence rates, and an extension to asynchronous stochastic gradient descent-like algorithms, such as SDCA.

## Footnotes

*The work is supported in part by the National Key R&D Program of China 2017YFB0202902, China Scholarship Council, NSF DMS-1720237, and ONR N000141712162

[2]Additionally, the step size needs to be modified to ensure convergence results hold. However in practice traditional step sizes appear to still allow convergence, barring extreme circumstances.

[3]The computing can start before the reading is completed. If $\nabla_i f(\hat{x})$ does not require all components of $\hat{x}$, only the required ones are read.

[4]say, because they are larger, bear more nonzero entries in the training set, or suffer poorer data locality.

[5]The distribution doesn't have to be uniform. We need only assume that every block has a nonzero probability of being updated. It is easy to adjust our analysis to this case.

[6]See, for example, (5.1) and (A.10) in [18], and (14) and Lemma 4 in [6].

[7]A function $f$ is coercive if $\|x\| \to \infty$ means $f(x) \to \infty$.

[8]A condition weaker than $\nu$-strong convexity and useful for problems involving an underdetermined linear mapping $Ax$; see [11, 13].

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
