[Supplementary Material]

## Supplementary Material

Our analysis uses the following standard inequalities. For any $x^1, x^2, \ldots, x^M \in \mathbb{R}^N$ and $\varepsilon > 0$, it holds that

$$\langle x^1, x^2 \rangle \leq \varepsilon \|x^1\|_2^2 + \frac{1}{\varepsilon}\|x^2\|_2^2 \tag{31}$$

$$\langle x^1, x^2 \rangle \leq \|x^1\|_2 \cdot \|x^2\|_2 \tag{32}$$

$$\|\sum_{i=1}^{M} x^i\|_2^2 \leq M \left( \sum_{i=1}^{M} \|x^i\|_2^2 \right) \tag{33}$$

$$\|d^k\|_2 = \|x^k - \hat{x}^k\|_2 \leq \sum_{i=k-j(k)}^{k-1} \|\Delta^i\|_2 \tag{34}$$

The last inequality is derived from (4), where $d^k$ is defined, using a telescoping sum and the triangle inequality.

### Proof of Lemma 1

Note that $\Delta_i^k = \delta(i, i_k) \cdot \Delta_{i_k}^k$, where $\delta(i, j)$ denotes the Kronecker delta: $\delta(i, j) = \left\{ \begin{array}{ll} 0, & i = j \\ 1, & \text{else} \end{array} \right.$.
Recalling the algorithm (2), we have:

$$-\langle \Delta^k, \nabla f(\hat{x}^k) \rangle = -\langle \Delta_{i_k}^k, \nabla_{i_k} f(\hat{x}^k) \rangle = \frac{L}{\gamma}\|\Delta^k\|_2^2. \tag{35}$$

Since $\nabla f$ is $L$-Lipschitz,

$$f(x^{k+1}) \leq f(x^k) + \langle \nabla f(x^k), \Delta^k \rangle + \frac{L}{2}\|\Delta^k\|_2^2. \tag{36}$$

Hence

$$f(x^{k+1}) - f(x^k) \overset{(35)(36)}{\leq} \langle \nabla f(x^k) - \nabla f(\hat{x}^k), \Delta^k \rangle + (\frac{L}{2} - \frac{L}{\gamma})\|\Delta^k\|_2^2$$

$$\overset{a)}{\leq} L\|d^k\|_2 \cdot \|\Delta^k\|_2 + (\frac{L}{2} - \frac{L}{\gamma})\|\Delta^k\|_2^2$$

$$\overset{(34), j(k) \leq \tau}{\leq} L \sum_{d=k-\tau}^{k-1} \|\Delta^d\|_2 \cdot \|\Delta^k\|_2 + (\frac{L}{2} - \frac{L}{\gamma})\|\Delta^k\|_2^2$$

$$\overset{b)}{\leq} \frac{L}{2\varepsilon} \sum_{i=k-\tau}^{k-1} \|\Delta^i\|_2^2 + \left[ \frac{(\tau\varepsilon+1)L}{2} - \frac{L}{\gamma} \right] \|\Delta^k\|_2^2, \tag{37}$$

where a) follows from (32) and the Lipschitzness of $\nabla f$, and b) is obtained by applying $a \cdot b \leq \frac{1}{2\varepsilon}|a|^2 + \frac{1}{2\varepsilon}|b|^2$ (where $\varepsilon > 0$ is arbitrary) to each term in the sum.

If $\gamma < \frac{2}{2\tau+1}$, we can choose $\varepsilon > 0$ such that $\varepsilon + \frac{1}{\varepsilon} = 1 + \frac{1}{\tau}(\frac{1}{\gamma} - \frac{1}{2})$. Then, it can be verified by direct calculation and substitutions that we have:

$$\xi_k - \xi_{k+1} \overset{(10)}{=} f(x^k) - f(x^{k+1}) + \frac{L}{2\varepsilon} \sum_{i=k-\tau}^{k-1} (i - (k-\tau) + 1)\|\Delta^i\|_2^2$$

$$- \frac{L}{2\varepsilon} \sum_{i=k+1-\tau}^{k-1} (i - (k-\tau))\|\Delta^i\|_2^2 - \frac{L}{2\varepsilon}\tau\|\Delta^k\|_2^2$$

$$\overset{c)}{=} f(x^k) - f(x^{k+1}) + \frac{L}{2\varepsilon} \sum_{i=k-\tau}^{k-1} \|\Delta^i\|_2^2 - \frac{L}{2\varepsilon}\tau\|\Delta^k\|_2^2 \overset{(37)}{\geq} \frac{1}{2}(\frac{1}{\gamma} - \frac{1}{2} - \tau)L \cdot \|\Delta^k\|_2^2, \tag{38}$$

where c) follows from $(i - (k-\tau) + 1)\|\Delta^i\|_2^2 - (i - (k-\tau))\|\Delta^i\|_2^2 = \|\Delta^i\|_2^2$. Therefore we have $\|\Delta^k\|_2^2 \in \ell^1$ by using a telescoping sum[8]. This immediately implies (12), and (13) follows from [Lemma 3, [5]].

**Proof of Theorem 1**

Let $t = t(k) = \lfloor k/N' \rfloor$. Recall $K(i,t)$ is defined at Sec. 1.1. Notice we have:

$$\|\nabla_i f(x^k)\|_2 \overset{a)}{\leq} \|\nabla_i f(\hat{x}^{K(i,t)})\|_2 + \|\nabla_i f(x^k) - \nabla_i f(\hat{x}^k)\|_2 + \|\nabla_i f(\hat{x}^k) - \nabla_i f(\hat{x}^{K(i,t)})\|_2$$

$$\overset{b)}{\leq} \|\nabla_i f(\hat{x}^{K(i,t)})\|_2 + L\|d^k\|_2 + L \sum_{j=K(i,t)}^{k-1} \|\hat{x}^{j+1} - \hat{x}^j\|_2, \tag{39}$$

where a) is by the triangle inequality and b) by Lipschitzness of $\nabla f$ and then applying the triangle inequality to the expansion of $\|\hat{x}^k - \hat{x}^{K(i,t)}\|$. We now bound each of the right-hand terms.

From Lemma 1 and by (34), we have

$$\lim_k \|d^k\|_2 \leq \lim_k \sum_{i=k-\tau}^{k-1} \|\Delta^i\|_2 = 0, \tag{40}$$

since $\Delta^k \to 0$. By the triangle inequality, we can derive

$$\|\hat{x}^{k+1} - \hat{x}^k\|_2 \leq \|d^k\|_2 + \|d^{k+1}\|_2 + \|\Delta^k\|_2. \tag{41}$$

Taking the limit,

$$\lim_k \|\hat{x}^{k+1} - \hat{x}^k\|_2 = 0. \tag{42}$$

Now notice:

$$\|\nabla_i f(\hat{x}^{K(i,t)})\|_2 = \|\nabla_{i_{K(i,t)}} f(\hat{x}^{K(i,t)})\|_2 = \frac{L}{\gamma} \|\Delta^{K(i,t)}\|_2. \tag{43}$$

Since, as $k \to \infty$, $K(i,t) \to \infty$ and $\|\Delta^k\|_2 \to 0$, this last term converges to 0 and the limit result is proven. The running best rate is obtained through the following argument: since $\|\Delta^k\|_2$ is square summable (by Lemma 1), so are $\|d^k\|_2$ by (34), $\|\hat{x}^{k+1} - \hat{x}^k\|_2$ by (41), and $\|\nabla_i f(\hat{x}^{K(i,t)})\|_2$ (since $t = \Theta(k)$) by (43). Hence, $\|\nabla_i f(x^k)\|_2$ is square summable. This implies $\|\nabla f(x^k)\|_2$ is square summable, hence $\lim_k \|\nabla f(x^k)\|_2 = 0$, and we obtain the running best rate again from [Lemma 3, [5]].

**Proof of Theorem 2**

Taking the expectation on both sides of (15) and multiplying $N$ yields

$$N\mathbb{E}\|\nabla_{i_k} f(x^{k-\tau})\|_2 = \sum_{i=1}^{N} \mathbb{E}\|\nabla_i f(x^{k-\tau})\|_2. \tag{44}$$

By $\|\cdot\|_2 \leq \|\cdot\|_1$, we obtain:

$$\mathbb{E}\|\nabla f(x^{k-\tau})\|_2 \leq \sum_{i=1}^{N} \mathbb{E}\|\nabla_i f(x^{k-\tau})\|_2 \overset{(44)}{=} N\mathbb{E}\|\nabla_{i_k} f(x^{k-\tau})\|_2. \tag{45}$$

In the next part, we prove $\mathbb{E}\|\nabla_{i_k} f(x^{k-\tau})\|_2 \to 0$. From (11), we can see that $(\|\Delta^k\|_2)_{k\geq 0}$ is bounded. The dominated convergence theorem implies:

$$\lim_k \mathbb{E}\|\Delta^k\|_2 = 0. \tag{46}$$

By (34), we have:

$$\lim_k \mathbb{E}(\|d^k\|_2) = 0. \tag{47}$$

Hence,

$$\lim_k \mathbb{E}\|\nabla_{i_k} f(\hat{x}^k)\|_2 \overset{(2)}{=} \frac{L}{\gamma} \lim_k \mathbb{E}\|\Delta^k\|_2 = 0. \tag{48}$$

The triangle inequality and $L$-Lipschitz continuity yield

$$\mathbb{E}\|\nabla_{i_k} f(x^{k-\tau})\|_2 \leq \mathbb{E}\|\nabla_{i_k} f(\hat{x}^k)\|_2 + \mathbb{E}\|\nabla_{i_k} f(x^k) - \nabla_{i_k} f(\hat{x}^k)\|_2$$
$$+ \mathbb{E}\|\nabla_{i_k} f(x^k) - \nabla_{i_k} f(x^{k-\tau})\|_2$$
$$\leq \mathbb{E}\|\nabla_{i_k} f(\hat{x}^k)\|_2 + L \cdot \mathbb{E}\|d^k\|_2 + L \sum_{i=k-\tau}^{k-1} \mathbb{E}\|\Delta^i\|_2. \qquad (49)$$

Applying (46), (47), and (48) to (49) yields

$$\lim_k \mathbb{E}\|\nabla_{i_k} f(x^{k-\tau})\|_2 = 0. \qquad (50)$$

Using (45), (50) yields

$$\lim_k \mathbb{E}\|\nabla f(x^{k-\tau})\|_2 = 0, \qquad (51)$$

which is equivalent to

$$\lim_k \mathbb{E}\|\nabla f(x^k)\|_2 = 0. \qquad (52)$$

Following a proof similar to that of Theorem 1 (except with added expectations), $\mathbb{E}\|\nabla f(x^k)\|_2^2$ is summable and thus has the running best rate $\mathcal{O}(1/k)$.

**Proof of Lemma 2**

The proof consists of two steps: in the first one, we prove

$$\pi_k - \pi_{k+1} \geq \frac{L}{4\tau}(\frac{1}{\gamma} - \frac{1}{2} - \tau) \cdot (\mathbb{E}S(k+1, \tau+1)), \qquad (53)$$

while in the second one, we prove

$$\pi_k^2 \leq \beta \cdot (\mathbb{E}S(k+1, \tau+1)) \cdot (\delta\tau\mathbb{E}S(k, \tau) + \mathbb{E}\|x^k - \overline{x^k}\|_2^2). \qquad (54)$$

Combining (53) and (54) gives us the claim in the lemma.

**Proving (53):** Since $\gamma < \frac{2}{2\tau+1}$, we can choose $\varepsilon > 0$ such that

$$\varepsilon + \frac{1}{\varepsilon} = 1 + \frac{1}{\tau}(\frac{1}{\gamma} - \frac{1}{2}) \qquad (55)$$

Direct subtraction of $F_k$ and $F_{k+1}$ yields:

$$F_k - F_{k+1} \overset{a)}{\geq} f(x^k) - f(x^{k+1}) + \delta \sum_{i=k-\tau}^{k-1} (i - (k-\tau) + 1)\|\Delta^i\|_2^2$$

$$- \delta \sum_{i=k+1-\tau}^{k-1} (i - (k-\tau))\|\Delta^i\|_2^2 - \delta\tau\|\Delta^k\|_2^2$$

$$\overset{b)}{=} f(x^k) - f(x^{k+1}) + \delta S(k, \tau) - \delta\tau\|\Delta^k\|_2$$

$$\overset{c)}{\geq} (\delta - \frac{L}{2\varepsilon})S(k, \tau) + \left[\frac{L}{\gamma} - \frac{(\tau\varepsilon+1)L}{2} - \delta\tau\right]\|\Delta^k\|_2^2$$

$$\overset{d)}{=} \frac{L}{4\tau}(\frac{1}{\gamma} - \frac{1}{2} - \tau) \cdot S(k, \tau) + \frac{L}{4}(\frac{1}{\gamma} - \frac{1}{2} - \tau) \cdot \|\Delta^k\|_2^2$$

$$\overset{e)}{\geq} \frac{L}{4\tau}(\frac{1}{\gamma} - \frac{1}{2} - \tau) \cdot S(k, \tau) + \frac{L}{4\tau}(\frac{1}{\gamma} - \frac{1}{2} - \tau) \cdot \|\Delta^k\|_2^2$$

$$\overset{f)}{=} \frac{L}{4\tau}(\frac{1}{\gamma} - \frac{1}{2} - \tau) \cdot S(k+1, \tau+1), \qquad (56)$$

where a) follows from the definition $F_k$, b) from the definition of $S(k, \tau)$, c) from (37), d) is a direct computation using (55), e) is due to $\tau \geq 1$, and f) is also a result of the definition of $S(k, \tau)$.

**Proving (54):** The convexity of $f$ yields

$$f(x^k) - f(\overline{x^k}) \leq \langle \nabla f(x^k), \overline{x^k} - x^k \rangle. \tag{57}$$

Let

$$a^k := \begin{pmatrix} \overline{x^k} - x^k \\ \sqrt{\delta\tau}\Delta^{k-1} \\ \vdots \\ \sqrt{\delta\tau}\Delta^{k-\tau} \end{pmatrix}, \quad b^k := \begin{pmatrix} \nabla f(x^k) \\ \sqrt{\delta\tau}\Delta^{k-1} \\ \vdots \\ \sqrt{\delta\tau}\Delta^{k-\tau} \end{pmatrix}. \tag{58}$$

Using this and the definition of $F_k$ (18), we have:

$$F_k - \min f \leq \langle a^k, b^k \rangle \leq \|a^k\|_2 \|b^k\|_2. \tag{59}$$

We bound $\mathbb{E}\|\nabla_{i_k} f(x^{k-\tau})\|_2^2$ as follows:

$$\mathbb{E}\|\nabla_{i_k} f(x^{k-\tau})\|_2^2 \overset{a)}{\leq} \mathbb{E}\big(\|\nabla_{i_k} f(x^k)\|_2 + \|\nabla_{i_k} f(x^{k-\tau}) - \nabla_{i_k} f(x^k)\|_2\big)^2$$

$$\overset{b)}{\leq} 2\mathbb{E}\|\nabla_{i_k} f(x^k)\|_2^2 + 2L^2\tau \sum_{i=k-\tau}^{k-1} \mathbb{E}\|\Delta^i\|_2^2$$

$$\overset{c)}{\leq} 4\mathbb{E}\|\nabla_{i_k} f(\hat{x}^k)\|_2^2 + 4L^2\mathbb{E}\|d^k\|_2^2 + 2L^2\tau \sum_{i=k-\tau}^{k-1} \mathbb{E}\|\Delta^i\|_2^2$$

$$= \tfrac{4L^2}{\gamma^2}\mathbb{E}\|\Delta^k\|_2^2 + 6L^2\tau \sum_{i=k-\tau}^{k-1} \mathbb{E}\|\Delta^i\|_2^2, \tag{60}$$

where a) follows from the triangle inequality, b) from the Lipschitzness of $\nabla f$ and (33), and c) from $\|\nabla_{i_k} f(x^k)\|_2^2 \leq 2\|\nabla_{i_k} f(\hat{x}^k)\|_2^2 + 2\|d^k\|_2^2$ and (34). We also have the bound

$$\|\nabla f(x^k)\|_2^2 \leq 2\|\nabla f(x^{k-\tau})\|_2^2 + 2L^2\tau \sum_{i=k-\tau}^{k-1} \|\Delta^i\|_2^2, \tag{61}$$

Hence, applying (44) to (60) yields

$$\mathbb{E}\|\nabla f(x^{k-\tau})\|_2^2 \leq \tfrac{4NL^2}{\gamma^2}\mathbb{E}\|\Delta^k\|_2^2 + 6NL^2\tau \sum_{i=k-\tau}^{k-1} \mathbb{E}\|\Delta^i\|_2^2,$$

and further with (61),

$$\mathbb{E}\|\nabla f(x^k)\|_2^2 \leq \tfrac{8NL^2}{\gamma^2}\mathbb{E}\|\Delta^k\|_2^2 + (12N+2)L^2\tau \sum_{i=k-\tau}^{k-1} \mathbb{E}\|\Delta^i\|_2^2. \tag{62}$$

Finally we obtain (54) from

$$\pi_k^2 = [\mathbb{E}(F_k - \min f)]^2 \overset{(59)}{\leq} \mathbb{E}(\|a^k\|_2\|b^k\|_2)^2 \leq \mathbb{E}(\|a^k\|_2^2) \cdot \mathbb{E}(\|b^k\|_2^2)$$

$$\overset{a)}{\leq} (\delta\tau\mathbb{E}S(k,\tau) + \mathbb{E}\|\nabla f(x^k)\|_2^2) \times (\delta\tau\mathbb{E}S(k,\tau) + \mathbb{E}\|x^k - \overline{x^k}\|_2^2)$$

$$\overset{b)}{\leq} \beta\mathbb{E}S(k+1,\tau+1) \cdot (\delta\tau\mathbb{E}S(k,\tau) + \mathbb{E}\|x^k - \overline{x^k}\|_2^2), \tag{63}$$

where a) follows from the definitions of $a^k, b^k$ and b) from (62) and the definition of $S(k,\tau)$.

**Proof of Theorem 3**

With (56), we can see that $f(x^k) \leq F_k \leq F_0$. Since $f$ is coercive, the sequence $(x^k)_{k\geq 0}$ is bounded. Hence, we have $\sup_k\{\|x^k - \overline{x^k}\|_2\} < +\infty$. Hence, there exists $R > 0$ such that

$$\alpha\big(\sum_{i=k-\tau}^{k-1} \tau\delta\mathbb{E}\|\Delta^i\|_2^2 + \mathbb{E}\|x^k - \overline{x^k}\|_2^2\big) \leq \tfrac{1}{R}. \tag{64}$$

for all $k$. Using Lemma 2, we have

$$\pi_k - \pi_{k+1} \geq R\pi_k^2. \tag{65}$$

Using (56), we can see that $\pi_k \geq \pi_{k+1}$ for all $k$. Thus, we have

$$\pi_k - \pi_{k+1} \geq R\pi_{k+1}\pi_k \tag{66}$$

$$\implies \frac{1}{\pi_{k+1}} - \frac{1}{\pi_k} \geq R. \tag{67}$$

Therefore, using a telescoping sum, we can deduce that:

$$\pi_{k+1} \leq \frac{1}{kR + \frac{1}{\pi_0}}. \tag{68}$$

Noting $\mathbb{E}(f(x^k) - \min f) \leq \pi_k$, we have proven the result.

**Proof of Theorem 4**

We have

$$\mathbb{E}(f(x^k) - \min f) \geq \nu\mathbb{E}\|x^k - \overline{x^k}\|_2^2, \tag{69}$$

Hence recalling the definition from (18), we have

$$\mathbb{E}\pi_k \geq \nu\mathbb{E}\|x^k - \overline{x^k}\|_2^2 + \sum_{i=k-\tau}^{k-1} \delta\mathbb{E}\|\Delta^i\|_2^2 \geq \min\{\nu, 1\}(\mathbb{E}\|x^k - \overline{x^k}\|_2^2 + S(k, \tau)).$$

Using this, the monotonicity of $\pi^k$, and Lemma 2 yields

$$\pi_k\pi_{k+1} \leq (\pi_k)^2 \leq \frac{\alpha}{\min\{\nu,1\}}(\pi_k - \pi_{k+1}) \cdot \pi_k. \tag{70}$$

Rearranging this yields the result.

**Proof of Lemma 3**

The Lipschitz continuity of $\nabla f$ yields

$$f(x^{k+1}) - f(x^k) \leq \langle \nabla f(x^k), \Delta^k \rangle + \frac{L}{2}\|\Delta^k\|_2^2$$

$$\overset{a)}{=} \langle \nabla f(x^k) - \nabla f(\hat{x}^k), \Delta^k \rangle + (\frac{L}{2} - \frac{L}{\gamma})\|\Delta^k\|_2^2$$

$$\leq L\|d^k\|_2 \cdot \|\Delta^k\|_2 + (\frac{L}{2} - \frac{L}{\gamma})\|\Delta^k\|_2^2, \tag{71}$$

where a) is from $-\frac{L}{\gamma}\|\Delta^k\|_2^2 = \langle \nabla f(\hat{x}^k), \Delta^k \rangle$. We bound the expectation of $\|d^k\|_2^2$ over the delay and using (33), we have:

$$\mathbb{E}_{\vec{j}(k)}\big(\|d^k\|_2^2 \mid \chi^k\big) \leq \mathbb{E}_{\vec{j}(k)}\Big(\sum_{l=1}^{j(k)} j(k)\|\Delta^{k-l}\|_2^2 \mid \chi^k\Big)$$

$$\leq \sum_{j=1}^{+\infty} jp_j \sum_{l=1}^{j} \|\Delta^{k-l}\|_2^2 \overset{b)}{=} \sum_{l=1}^{+\infty}(\sum_{j=l}^{+\infty} jp_j)\|\Delta^{k-l}\|_2^2 \overset{c)}{\leq} \sum_{i=0}^{k-1} c_{k-i}\|\Delta^i\|_2^2, \tag{72}$$

where in b), we switched the order of summation in the double sum, and c) uses $\sum_{j=l}^{+\infty} jp_j \leq c_l$. Taking total expectation $\mathbb{E}(\cdot)$ on both sides of (72), we obtain

$$\mathbb{E}\|d^k\|_2^2 \leq \sum_{i=0}^{k-1} c_{k-i}\mathbb{E}\|\Delta^i\|_2^2 \overset{d)}{\leq} \sum_{i=0}^{k-1} c_{k-1-i}\mathbb{E}\|\Delta^i\|_2^2 = R(k-1), \tag{73}$$

where d) is by the fact $(c_i)_{i\geq 0}$ is descending. Hence:

$$\mathbb{E}[f(x^{k+1}) - f(x^k)] \leq L\mathbb{E}\|d^k\|_2 \cdot \|\Delta^k\|_2 + (\frac{L}{2} - \frac{L}{\gamma})\mathbb{E}\|\Delta^k\|_2^2$$

$$\leq \frac{L}{2\varepsilon}\mathbb{E}\|d^k\|_2^2 + \left[\frac{(\varepsilon+1)L}{2} - \frac{L}{\gamma}\right]\mathbb{E}\|\Delta^k\|_2^2$$

$$\leq \frac{L}{2\varepsilon}\sum_{l=1}^{+\infty}(\sum_{j=l}^{+\infty} jp_j)\mathbb{E}\|\Delta^{k-l}\|_2^2 + \left[\frac{(\varepsilon+1)L}{2} - \frac{L}{\gamma}\right]\mathbb{E}\|\Delta^k\|_2^2. \tag{74}$$

Since $\gamma < \frac{2}{2\sqrt{c_0}+1}$, we can choose $\varepsilon > 0$ such that

$$\tfrac{1}{2}(\varepsilon + \tfrac{c_0}{\varepsilon}) = \tfrac{1}{\gamma} - \tfrac{1}{2}. \tag{75}$$

With such $\varepsilon$ and (74), direct calculation using the definition of $G^k$ yields (22). When $\gamma < \frac{2}{2\sqrt{c_0}+1}$, $\frac{L}{2}(\frac{1}{\gamma} - \frac{1}{2} - \sqrt{c_0}) > 0$. From (22), we can see $(R(k))_{k \geq 0}$ is summable (telescoping sum). Thus, we have $\lim_k R(k) = 0$. Then note (73) and

$$c_0 \mathbb{E}\|\Delta^k\|_2^2 \leq \sum_{i=0}^{k} c_{k-i}\mathbb{E}(\|\Delta^i\|_2^2) = R(k). \tag{76}$$

Hence then have

$$\lim_k \mathbb{E}(\|d^k\|_2^2) = 0, \quad \lim_k \mathbb{E}(\|\Delta^k\|_2^2) = 0. \tag{77}$$

**Proof of Theorem 5**

Let $t = t(k) = \lfloor k/N' \rfloor$. Recalling $K(i,t)$ is defined at Sec. 1.1, we have:

$$\|\nabla_i f(x^k)\|_2 \overset{a)}{\leq} \|\nabla_i f(\hat{x}^{K(i,t)})\|_2 + \|\nabla_i f(x^{K(i,t)}) - \nabla_i f(\hat{x}^{K(i,t)})\|_2 + \|\nabla_i f(x^k) - \nabla_i f(x^{K(i,t)})\|_2$$

$$\overset{b)}{\leq} \|\nabla_i f(\hat{x}^{K(i,t)})\|_2 + L\|d^{K(i,t)}\|_2 + L \sum_{j=K(i,t)}^{k-1} \|\Delta^j\|_2, \tag{78}$$

where a) is by the triangle inequality and b) by the Lipschitzness of $\nabla f$ and then applying the triangle inequality to the expansion of $\|x^k - x^{K(i,t)}\|$. We now bound each of the right-hand terms.

Since, as $k \to \infty$, $K(i,t) \to \infty$. With the Cauchy-Schwarz inequality and (23), we have

$$\lim_k \mathbb{E}\|d^{K(i,t)}\|_2 \leq \lim_k (\mathbb{E}\|d^{K(i,t)}\|_2^2)^{\frac{1}{2}} = 0. \tag{79}$$

By $\lim_j \mathbb{E}\|\Delta^j\|_2 \leq \lim_j (\mathbb{E}\|\Delta^j\|_2^2)^{\frac{1}{2}} = 0$,

$$\lim_k L \sum_{j=K(i,t)}^{k-1} \mathbb{E}\|\Delta^j\|_2 = 0. \tag{80}$$

Now notice:

$$\|\nabla_i f(\hat{x}^{K(i,t)})\|_2 = \|\nabla_{i_{K(i,t)}} f(\hat{x}^{K(i,t)})\|_2 = \frac{L}{\gamma}\|d^{K(i,t)}\|_2. \tag{81}$$

Since $\mathbb{E}\|d^{K(i,t)}\|_2 \to 0$ as $K(i,t) \to \infty$, we have

$$\lim_k \mathbb{E}\|\nabla_i f(\hat{x}^{K(i,t)})\|_2 = 0. \tag{82}$$

Taking expectations on both sides of (78), and using (79), (80) and (82), we then prove the result.

**Proof of Theorem 6**

Recall $j(k)$ defined near (3). Similar to the bound of $\|d^k\|_2^2$ in (34), we have

$$\mathbb{E}_{\vec{j}(k)}\big(\|x^k - x^{k-j(k)}\|_2^2 \mid \chi^k\big) \leq \sum_{i=0}^{k-1} s_{k-1-i}\|\Delta^i\|_2^2. \tag{83}$$

Taking total expectations of both sides yields

$$\mathbb{E}\|x^k - x^{k-j(k)}\|_2^2 \leq \sum_{i=0}^{k-1} s_{k-1-i}\mathbb{E}\|\Delta^i\|_2^2. \tag{84}$$

We have

$$\mathbb{E}\|\nabla_{i_k}f(x^{k-j(k)})\|_2^2 \overset{a)}{\leq} \mathbb{E}(\|\nabla_{i_k}f(x^k)\|_2 + \|\nabla_{i_k}f(x^{k-j(k)}) - \nabla_{i_k}f(x^k)\|_2)^2$$

$$\overset{b)}{\leq} 2\mathbb{E}\|\nabla_{i_k}f(x^k)\|_2^2 + 2L^2\mathbb{E}\|x^k - x^{k-j(k)}\|_2^2$$

$$\overset{c)}{\leq} 4\mathbb{E}\|\nabla_{i_k}f(\hat{x}^k)\|_2^2 + 4L^2\mathbb{E}\|d^k\|_2^2 + 2L^2\mathbb{E}\|x^k - x^{k-j(k)}\|_2^2$$

$$\overset{d)}{\leq} \frac{4L^2}{\gamma^2}\mathbb{E}\|\Delta^k\|_2^2 + 6L^2\sum_{i=0}^{k-1}s_{k-1-i}\mathbb{E}\|\Delta^i\|_2^2, \tag{85}$$

where a) follows from the triangle inequality, b) from the Lipschitzness of $\nabla f$ and (33), and c) from $\|\nabla_{i_k}f(x^k)\|_2^2 \leq 2\|\nabla_{i_k}f(\hat{x})\|_2^2 + 2L^2\|d^k\|_2^2$ and (34), and d) from (84). Taking total expectation of both sides of assumption (25) yields

$$\mathbb{E}\|\nabla_{i_k}f(x^{k-j(k)})\|_2^2 = \frac{\mathbb{E}\|\nabla f(x^{k-j(k)})\|_2^2}{N}. \tag{86}$$

By the triangle inequality,

$$\|\nabla f(x^k)\|_2^2 \leq 2\|\nabla f(x^{k-j(k)})\|_2^2 + 2L^2\sum_{i=0}^{k-1}s_{k-1-i}\mathbb{E}\|\Delta^i\|_2^2. \tag{87}$$

Hence, combining (86) and (87) produces

$$\mathbb{E}\|\nabla f(x^{k-j(k)})\|_2^2 \leq \frac{4NL^2}{\gamma^2}\mathbb{E}\|\Delta^k\|_2^2 + 6NL^2\sum_{i=0}^{k-1}s_{k-1-i}\mathbb{E}\|\Delta^i\|_2^2;$$

which is substituted into (87) to yield

$$\mathbb{E}\|\nabla f(x^k)\|_2^2 \leq \frac{8NL^2}{\gamma^2}\mathbb{E}\|\Delta^k\|_2^2 + (12N+2)L^2\sum_{i=0}^{k-1}s_{k-1-i}\mathbb{E}\|\Delta^i\|_2^2. \tag{88}$$

By $\sum_{i=0}^{k-1}s_{k-1-i} \leq \sum_{i=0}^{k-1}c_{k-1-i}\mathbb{E}\|\Delta^i\|_2^2 = R(k-1)$ and (22),

$$\lim_k \mathbb{E}\|\nabla f(x^k)\|_2^2 = 0. \tag{89}$$

The proof is completed by applying the Cauchy-Schwarz inequality

$$\mathbb{E}\|\nabla f(x^k)\|_2 \leq (\mathbb{E}\|\nabla f(x^k)\|_2^2)^{\frac{1}{2}}. \tag{90}$$

**Proof of Lemma 4**

This proof is very similar to Lemma 2 except that $R(k)$ plays the role of $S(k,\tau)$. Let

$$a^k = \begin{pmatrix} \overline{x^k} - x^k \\ \sqrt{c_0\overline{\delta}}\Delta^{k-1} \\ \vdots \\ \sqrt{c_k\overline{\delta}}\Delta^0 \end{pmatrix}, b^k = \begin{pmatrix} \nabla f(x^k) \\ \sqrt{c_0\overline{\delta}}\Delta^{k-1} \\ \vdots \\ \sqrt{c_k\overline{\delta}}\Delta^0 \end{pmatrix}. \tag{91}$$

Thus, we have

$$G_k - \min f \leq \langle a^k, b^k \rangle \leq \|a^k\|_2\|b^k\|_2. \tag{92}$$

By taking expectations, we get

$$\mathbb{E}(G_k - \min f) \leq \mathbb{E}(\|a^k\|_2\|b^k\|_2) \leq [\mathbb{E}\|a^k\|_2^2 \cdot \mathbb{E}\|b^k\|_2^2]^{1/2}. \tag{93}$$

By (88) and the definitions of $a^k, b^k, R(k)$, we get

$$[\mathbb{E}(G_k - \min f)]^2 \leq \mathbb{E}(\|a^k\|_2^2) \cdot \mathbb{E}(\|b^k\|_2^2)$$

$$\leq (\overline{\delta}R(k) + \mathbb{E}\|\nabla f(x^k)\|_2^2) \times (\overline{\delta}R(k) + \mathbb{E}\|x^k - \overline{x^k}\|_2^2)$$

$$\leq \overline{\beta}R(k) \times (R(k) + \mathbb{E}\|x^k - \overline{x^k}\|_2^2). \tag{94}$$

Finally, from the definition of $\overline{\alpha}$ and Lemma 3, the theorem follows.

**Proof of Theorem 7**

We have

$$\mathbb{E}(f(x^k) - \min f) \geq \nu \mathbb{E}\|x^k - \overline{x^k}\|_2^2, \tag{95}$$

which also means that

$$\mathbb{E}(\bar{\delta} R(k) + \|x^k - \overline{x^k}\|_2^2) \leq \max\{1, \frac{1}{\nu}\}\phi_k. \tag{96}$$

Lemma 4 yields

$$(\phi_k)^2 \leq \bar{\alpha} \max\{1, \frac{1}{\nu}\}(\phi_k - \phi_{k+1}) \cdot (\phi_k) \tag{97}$$

Note that $\phi_k$ is decreasing, we obtain

$$\phi_{k+1} \leq \bar{\alpha} \max\{1, \frac{1}{\nu}\}(\phi_k - \phi_{k+1}). \tag{98}$$

Then, we have the result by rearrangement.

**Proof of Lemma 5**

$$
\begin{aligned}
f(x^{k+1}) &\overset{a)}{\leq} f(x^k) + L\|d^k\|_2 \cdot \|\Delta^k\|_2 + (\frac{L}{2} - \frac{L}{\gamma_k})\|\Delta^k\|_2^2 \\
&\overset{b)}{\leq} f(x^k) + L\sum_{l=1}^{j(k)} \|\Delta^{k-l}\|_2 \cdot \|\Delta^k\|_2 + (\frac{L}{2} - \frac{L}{\gamma_k})\|\Delta^k\|_2^2 \\
&\overset{c)}{\leq} f(x^k) + L\sum_{l=1}^{j(k)} (\frac{\epsilon_l}{2}\|\Delta^{k-l}\|_2^2 + \frac{1}{2\epsilon_l}\|\Delta^k\|_2^2) + (\frac{L}{2} - \frac{L}{\gamma_k})\|\Delta^k\|_2^2 \\
&= f(x^k) + \frac{L}{2}\sum_{l=1}^{j(k)} \epsilon_l\|\Delta^{k-l}\|_2^2 + \frac{L}{2}\sum_{l=1}^{j(k)} \frac{1}{\epsilon_l}\|\Delta^k\|_2 + (\frac{L}{2} - \frac{L}{\gamma_k})\|\Delta^k\|_2^2 \\
&\overset{d)}{\leq} f(x^k) + \frac{L}{2}\sum_{l=1}^{+\infty} \epsilon_l\|\Delta^{k-l}\|_2^2 + \frac{L}{2}(1 + \sum_{l=1}^{j(k)} \frac{1}{\epsilon_l} - \frac{2}{\gamma_k})\|\Delta^k\|_2^2.
\end{aligned}
\tag{99}
$$

where a) follows from Lipschitzness of $\nabla f$ and definitions of $d^k, \Delta^k$, b) from the triangle inequality, c) from (32), and d) from $j(k) < \infty$. Then, a direct calculation yields the first result in (29). Hence the second follows by summability: $\|\Delta^k\|_2^2 \in \ell^1$.

$$\lim_{k \in Q_T} \|d^k\|_2 \leq \sum_{l=k-T}^{k-1} \lim_l \|\Delta^l\|_2 = 0 \tag{100}$$

$$L(\frac{1}{\gamma_k} - D_{j(k)})\|\Delta^k\|_2^2 = \frac{c(1-c)}{LD_{j(k)}}\|\nabla_{i_k} f(\hat{x}^k)\|_2^2. \tag{101}$$

Therefore,

$$\frac{1}{D_T}\sum_{k \in Q_T} \|\nabla_{i_k} f(\hat{x}^k)\|_2^2 < \sum_k \frac{\|\nabla_{i_k} f(\hat{x}^k)\|_2^2}{D_{j(k)}} < +\infty. \tag{102}$$

**Proof of Theorem 8**

For any $T$ and $k \in Q_T$, let $t = t(k) = \lfloor k/N' \rfloor$, and by the triangle inequality:

$$
\begin{aligned}
\|\nabla_i f(x^k)\| &\leq \|\nabla_i f(x^{K(i,t)}) - \nabla_i f(x^k)\|_2 \\
&\quad + \|\nabla_i f(\hat{x}^{K(i,t)}) - \nabla_i f(x^{K(i,t)})\|_2 + \|\nabla_i f(\hat{x}^{K(i,t)})\|_2.
\end{aligned}
\tag{103}
$$

From Lemma 5, we have

$$\lim_{k} \|\nabla_i f(x^{K(i,t)}) - \nabla_i f(x^k)\|_2 \leq \lim_{k} L \sum_{i=k-N'+1}^{k-1} \|\Delta^i\|_2 = 0. \tag{104}$$

Noting $K(i,t) \in Q_T$ by the ECSD assumption, we can derive

$$\lim_{k} \|\nabla_i f(\hat{x}^{K(i,t)}) - \nabla_i f(x^{K(i,t)})\|_2 \leq \lim_{k} L \|d^{K(i,t)}\|_2 = 0. \tag{105}$$

Now notice by Lemma 5:

$$\lim_{k} \|\nabla_i f(\hat{x}^{K(i,t)})\|_2 = \lim_{K(i,t)} \|\nabla_{i_{K(i,t)}} f(\hat{x}^{K(i,t)})\|_2 = 0.$$

Since $K(i,t) \to \infty$, this right term converges to 0 and the result is proven.

## Footnotes

[8]We write $a^k \in \ell^1$ if $\sum_{k=1}^{\infty} |a^k| < \infty$.