[Reviews · NeurIPS 2017]

Reviewer 1



This paper proves the convergence of Async-CD under weaker assumptions than existing work for deterministic/stochastic CD with bounded/unbounded delay for convex / nonconvex objectives. In particular, this paper does not assume the independence between the sample coordinate i and the staleness. Overall, this paper considers a more realistic scenario while using Async-CD, which is very interesting to me. I would like to adjust my score based on authors' response. Major comments: - How does the \epsilon in Lemma 1 affect the convergence? - Authors claim that the steplength gets improved in this paper. But I did not find the way how to choose it in Theorem 3. - The theorem for Async-CD/SCD omits the constant factor in the convergence rate result. Based on my own checking the proof, since the steplength chosen in this paper is bounded by 1/\tau, it will bring additional complexity O(\tau / \epsilon), which does not bring speedup. Can authors explain this? Typos: - line 78: it ensures *that*. Also, every block is updated *at least* the specified frequency here IMHO. - line 180: lim_k should be lim_{k->infty}. This notation needs to be changed throughout the paper. - line 313: the inequality for "ab <=" - Eq (9): the def of \Delta^i is missing - Eq (17): the def of \bar{x^k} is missing

Reviewer 2



The authors present an analysis of convergence for (asynchronous) BCD, under general assumptions. The paper is very well written, I really liked the way the authors introduced the topic, and presented their results. The description of the problem is very clear, the classical ODE analysis is also very well explained, and in general every step is carefully detailed. The proofs seem correct, the set of convergence results is important, and the techniques used (besides the classical Lyapunov functions) are quite interesting.

Reviewer 3



The paper studies coordinate descent with asynchronous updates. The key contributions include a relaxation of the assumptions on how the block coordinates are chosen and how large could the delays be. Descent lemma and convergence rates are obtained for a variety of settings using the ode analysis and discretization. I think the paper is well-grounded because existing assumptions in the asynchronous analyses of coordinate descents are highly unsatisfactory; and this work explained the problem well and proposed a somewhat more satisfactory solution. For this reason I vote for accept. Detailed evaluation: 1. I did not have time to check the proof, but I think the results are believable based on the steps that the authors outlined in the main paper. 2. The conditions for cyclic choice of coordinates and unbounded delays could be used in other settings. 3. In a related problem of asynchronous block coordinate frank wolfe, it was established under weaker tail bound assumptions of the delay random variable that there could be unbounded stochastic delay. The delay random variable also are not required to be independent. Establishing deterministic sequence in AP-BCFW or more heavy-tailed stochastic delay for AP-BCD could both be useful. - Wang, Y. X., Sadhanala, V., Dai, W., Neiswanger, W., Sra, S., & Xing, E. (2016, June). Parallel and distributed block-coordinate Frank-Wolfe algorithms. In International Conference on Machine Learning (pp. 1548-1557).

Reviewer 4



Paper proposes an analysis approach for asynchronous coordinate descent under, the authors claim, more realistic assumptions. Specifically, the approach does not require the assumptions of (1) independence of delays from block identity, and (2) blocks are chosen uniformly at random with replacement. The authors construct Lyapunov functions directly modelling the objective and delays, and prove convergence for non-convex functions, with additional rates for convex functions. The use of Lyanpov functions to directly model delays is, to the best of my knowledge, a new approach in literature. More-informed reviewers than me should evaluate on its novelty value. It is also not clear if the approach can be easily generalized and applied to other stochastic optimization algorithms. I do think that the paper does not give sufficient credit to existing literature on ridding the assumption of independence of delay with update identity. For example, Mania et al [12] addressed this issue by decoupling the parameters read by each core from the virtual parameters (which may never exist in consistent state) on which progress is actually defined. The work of Leblond et al [9], which was addressed in related work, was based on ideas of [14]. I also do not agree with the authors' assessment that [9] requires the entire data to be read before computing an update. Possible typos: Line 96 -- should this be O(\tau^{-3})? Line 169 -- "we design our the energy"